**Data Availability Statement:** All relevant data are within the article and its Supporting information files.

**Funding:** The review was funded by National Institute for Health Research (NIHR) as part of

# Characteristics of alcohol recovery narratives: Systematic review and narrative synthesis

Mohsan Subhani[1,2]*, Usman Talat[3], Holly Knight[4], Joanne R. Morling[1,2,4], Katy A. Jones[5], Guruprasad P. Aithal[1,2], Stephen D. Ryder[1,2], Joy Llewellyn-Beardsley[6], Stefan Rennick-Egglestone[6]

1 Nottingham Digestive Diseases Biomedical Research Centre (NDDC), School of Medicine, University of Nottingham, Nottingham, United Kingdom, 2 NIHR Nottingham Biomedical Research Centre, Nottingham University Hospitals NHS Trust and the University of Nottingham, Nottingham, United Kingdom, 3 Alliant Manchester Business School, University of Manchester, Manchester, England, 4 Population and Lifespan Sciences, University of Nottingham, Nottingham, United Kingdom, 5 School of Medicine, Applied Psychology, University of Nottingham, Nottingham, United Kingdom, 6 School of Health Sciences, Institute of Mental Health, University of Nottingham, Nottingham, United Kingdom

* mohsan.subhani@nottingham.ac.uk

## Abstract

### Background and aims

Narratives of recovery from alcohol misuse have been analysed in a range of research studies. This paper aims to produce a conceptual framework describing the characteristics of alcohol misuse recovery narratives that are in the research literature, to inform the development of research, policy, and practice.

### Methods

Systematic review was conducted following PRISMA guidelines. Electronic searches of databases (Ovid MEDLINE, EMBASE, CINHAL, PsychInfo, AMED and SCOPUS), grey literature, and citation searches for included studies were conducted. Alcohol recovery narratives were defined as "first-person lived experience accounts, which includes elements of adversity, struggle, strength, success, and survival related to alcohol misuse, and refer to events or actions over a period of time". Frameworks were synthesised using a three-stage process. Sub-group analyses were conducted on studies presenting analyses of narratives with specific genders, ages, sexualities, ethnicities, and dual diagnosis. The review was prospectively registered (PROSPERO CRD42021235176).

### Results

32 studies were included (29 qualitative, 3 mixed-methods, 1055 participants, age range 17-82years, 52.6% male, 46.4% female). Most were conducted in the United States (n = 15) and Europe (n = 11). No included studies analysed recovery narratives from lower income countries. Treatment settings included Alcoholic Anonymous (n = 12 studies), other formal treatment, and 'natural recovery'. Eight principle narrative dimensions were identified (genre, identity, recovery setting, drinking trajectory, drinking behaviours, stages, spirituality and religion, and recovery experience) each with types and subtypes. All dimensions were

feasibility randomised control trial. Funding award ID: NIHR201146. The funders had no role in study design, data collection and analysis, decision to publish, or preparation of the manuscript.

**Competing interests:** NO authors have competing interests.

present in most subgroups. *Shame* was a prominent theme for female narrators, *lack of sense of belonging* and *spirituality* were prominent for LGBTQ+ narrators, and *alienation* and *inequality* were prominent for indigenous narrators.

## Conclusions

Review provides characteristics of alcohol recovery narratives, with implications for both research and healthcare practice. It demonstrated knowledge gaps in relation to alcohol recovery narratives of people living in lower income countries, or those who recovered outside of mainstream services.

## Protocol registration

Prospero registration number: CRD42020164185.

## Introduction

Alcohol misuse [1] has been a cause of major public health concern. Globally over 2.3 billion people are current alcohol drinkers, and of these approximately 240 million are alcohol dependent [2]. In the United Kingdom (UK) 25% of the population drinks above the recommended level and 10% are harmful drinkers [3]. The UK has observed a 400% rise in mortality due to liver disease over the last three decades, and in 2020 Public Health England reported that alcohol specific deaths reached their highest since 2001 [3–5]. The estimated cost to the National Health Service to treat alcohol related problems is £3.5 billion annually and alcohol use contributes to over 200 different medical conditions [2, 3]. This emphasizes the importance of successful recovery from alcohol to misuse to minimise the associated harm.

Recovery from substance misuse has been described by the United Kingdom Drug Policy commission as a process of voluntarily controlling substance misuse aimed at maximising health and personal wellbeing benefits and social responsibility [6]. Although recovery from alcohol misuse is possible, and researchers have demonstrated successful models [7–9], little remains known at the individual level regarding recovery characteristics and related dimensions. In this context the notion of narrative psychology can contribute to a better understanding of recovery. Sarbin (1986); draws attention to narrative psychology, as "storied nature of human conduct"(1986); discussing how humans use stories to create meaning and share life experiences [10]. Bruner (1986) further argued there are two modes of thought and cognitive function: the paradigmatic and narrative modes. In the paradigmatic mode thoughts are presented as logical argument, whereas in narrative mode, as stories of particular events [11, 12].

Recovery narratives can be defined as personal stories of health problems and of recovery [13], which can be shared with others [14], and which can provide recipients with insights into the phenomenology of recovery [15]. In this regards, the Social Identity Model of Recovery (SIMOR) identifies alcohol recovery as "a process of social identity transitioning, wherein an individual becomes a member of a recovery-orientated group, and in so doing internalizes the values and beliefs of the in-group which, in turn, leads to a new sense of self (or recovery identity) that strongly guides their attitudes and behaviours" (page 113) [7, 16]. The act of sharing alcohol narratives has been an important component of the Alcohol Anonymous (AA) 12-step programme [17].

Narrative approaches to research have been broadly applied in health research [18–20], where they "allow for the intimate and in-depth study of the individual's experiences over time and in context" [21]. For example, recovery in people with stroke was facilitated by identity transformation using a metaphor of change in physical functioning and self-identity [22]. In another study sharing cancer stories and narratives of illness helped cancer patient to make choices and enabled a sense of belonging to a group [23]. Moreover, recovery narratives have been used to promote and encourage engagement with health services [24], where they might be used to extend clinical practice, including as a resource for people who are finding recovery challenging [25].

People have diverse experiences of both alcohol misuse and recovery [26] which may interact with their personal characteristics to influence choice of treatment. For example, the AA approach involves acceptance of an "AA identity" as an "alcoholic" and an experience of "hitting bottom", which enables participants to engage with support groups [27]. In comparison the narrative of the "self-changer" describes excessive, but not problematic, drinking and strong individual willpower to stop drinking [26]. Recovery from addiction is a dynamic process, it can follow a nonlinear pathway, and a successful recoveree may have interacted with more than one service or recovery strategy in their journey [28]. Once a person shares their lived experience as a narrative it can be processed in different ways by recipients (researcher, care provider, and patient) a phenomenon described as 'polysemy' by Bruner (1986) [12]. This in turn can introduce further complexity, that might affect intended use of narratives. This emphasises the importance of having a standardised framework in the field to describe characteristics of narratives. Indeed, the recovery narrative is an evolving concept in the field of drug and alcohol misuse, and has been a focus of discussion in contemporary literature [29]. Alcohol misuse recovery narratives have been studied by researchers to understand different processes of change [11], how people can recover in both the presence or absence of treatment [16], and how people differ on individual factors e.g., age, gender, ethnicity in recovery process [30]. Although alcohol misuse recovery narratives have been widely studied by the research community, no overarching conceptual framework for alcohol recovery narratives exists.

A recent systematic review synthesised evidence on the characteristics of mental health recovery narratives and generated a framework to describe how these narratives have been conceptualised by the research community [31]. The framework identified nine dimensions: genre, positioning, emotional tone, relationship with recovery, trajectory, use of turning points, narrative sequence, protagonists, and use of metaphor. Dimensions such as genre, relationship with recovery, turning points, and trajectory can be applicable to narratives of recovery from a range of other health conditions including alcohol misuse.

The aim of this review is to develop a conceptual framework describing the characteristics of alcohol recovery narratives that have been reported in the research literature. Benefits of producing this framework include: identifying gaps in knowledge e.g., narratives or narrators who have not been considered in research analyses, summarising the range of methods that have been used to collect and analyse narratives to date, understanding potential biases of these methods, informing content of educational courses that support people in sharing a narrative as a part of the recovery process [32], and enabling collective approaches that draw on sets of narrative knowledge to influence the health system.

## Methods

A systematic review and narrative synthesis was conducted following Preferred Reporting Items for Systematic Reviews and Meta-Analyses (PRISMA) guidance [33]. The protocol was prospectively registered with the Prospective Register of Systematic Reviews (PROSPERO

2021 CRD42021235176). The systematic review was conducted as part of the KLIFAD (Does knowledge of liver fibrosis affect high risk drinking behaviour?) study (UK National Institute for Health Research Research for Patient Benefit grant, NIHR201146).

The review team included researchers with specialist background in mental health, lived experience narratives [13], psychology, public health, epidemiology, qualitative research, alcohol care, and liver medicine.

## Eligibility criteria

Alcohol misuse recovery narratives were defined as "first-person lived experience accounts, which include elements of adversity, struggle, strength, success, and survival related to alcohol misuse, and refer to events or actions over a period of time". This modified a definition of mental health recovery narratives in the study by Llewellyn-Beardsley et al. [31].

**Inclusion criteria.**

- The study presents or substantially advance an original framework of typologies and/or themes of alcohol misuse recovery narratives.

- The framework is produced through an analysis of empirical data.

**Exclusion criteria.**

- The study is of narratives, but it is not possible to identify from title or abstract whether they are alcohol misuse recovery narratives

- The study is of narratives where the narrator does not have personal experience of alcohol misuse (for example the narratives are of family members of people who have misused alcohol).

**Primary outcome** was to develop a framework of over-arching narrative typologies (structures) and themes (content) characterizing alcohol recovery narratives which can be used by alcohol misuse support services to inform the development of future research, policy, and practice within healthcare and other settings.

**Secondary outcome** was to describe alcohol recovery narratives based on narrator's age, gender, sexuality, and ethnicity.

## Search strategy

A search strategy was designed in consultation with an expert librarian from University of Nottingham. Publication database searches was conducted using Ovid MEDLINE, EMBASE, CINHAL, PsychInfo, and AMED. A grey literature search was conducted using ProQuest, SCOPUS, and ClinicalTrials.gov. All searches were from inception to March 2021, and a backwards citation search was conducted by examining the reference list in each included publication. A sample search from Ovid Medline is provided in S1 Table, which was specialised to each database.

## Screening and data abstraction

Two reviewers (MS and UT) independently screened titles and abstracts for eligibility. A candidate list of included studies was crosschecked by both reviewers, along with a randomly selected 10% of excluded studies. Any conflicts in study inclusion were resolved through discussion with three further reviewers (SRE, KJ and JLB). Rayyan-QRCI systematic review software, Endnote (Version-X9) and Microsoft Excel were used to screen, remove duplicate entries, and record reviewers' decisions.

A Data Abstraction Table was designed and piloted. Three reviewers (MS, UT, and HK) extracted data from the included studies. The DAT included information about the lead

author, academic discipline, country of study, participant demographics (age, gender, country), study design, how alcohol recovery stories were named and defined by the authors, key characteristics of the study and alcohol recovery narrative 'types' (as identified by study authors) was extracted.

### Risk of bias and quality assessment

Quality assessment during qualitative evidence synthesis has been a matter of debate for many decades [34]. Cochrane Qualitative and Implementation Methods Group recommendations are to use a tool that takes the multi-dimensional nature of qualitative evidence into account [34].

Guided by this perspective, the quality of included studies and risk of bias was assessed using the Critical Appraisals Skills Programme (CASP tool for qualitative research [35]. The CASP tool focuses on three domains: study design, validity of results, and generalisability. Each domain is assessed using a set of questions. Based on the response to these questions the studies were marked as low, medium, or high quality. Studies which provided satisfactory information in all domains were marked as high quality, with missing or unsatisfactory information in one domain as medium quality, and with missing or unsatisfactory information in two or more domains as low quality.

### Data synthesis

The following three-stage narrative synthesis approach was adopted, modified from Popay (2006) [31, 36].

- The lead author formed an initial conceptual framework presenting a preliminary synthesis of findings of included studies,

- The conceptual framework was reviewed by the authors, and relationships between entities in the framework were explored

- The robustness of the synthesis was assessed by conducting selected subgroup analyses

Information in subgroups was assimilated through an inductive thematic analysis of the content of included studies, which considered social, cultural and demographics aspects.

In producing the initial conceptual framework, concepts from included studies were organised into themes and sub-themes. Concepts were merged which were sufficiently similar. Higher-level themes were organised into a three-level framework of form, structure, and content, informed by narrative theory [37].

**Original author language.** Where possible the language used by original authors was preserved, while maintaining the clarity of synthesis of dimension and characteristics of alcohol recovery narratives. Where the terms "alcoholic" or "alcoholism" were used by the original authors to describe alcohol misuse, these have been retained.

The review group acknowledges the heterogeneity in language used to describe alcohol use, and the stigma associated with some commonly used terms, which itself can act as barrier to change. After thoughtful discussion between review group, we opted for the term 'alcohol misuse' to describe excess alcohol intake, harmful alcohol intake, drinking problems, alcohol dependence, and alcohol use disorder.

## Results

A total of 11,332 records were initially identified. After applying eligibility criteria 32 documents were included in the final narrative synthesis (Fig 1). Most studies described in these

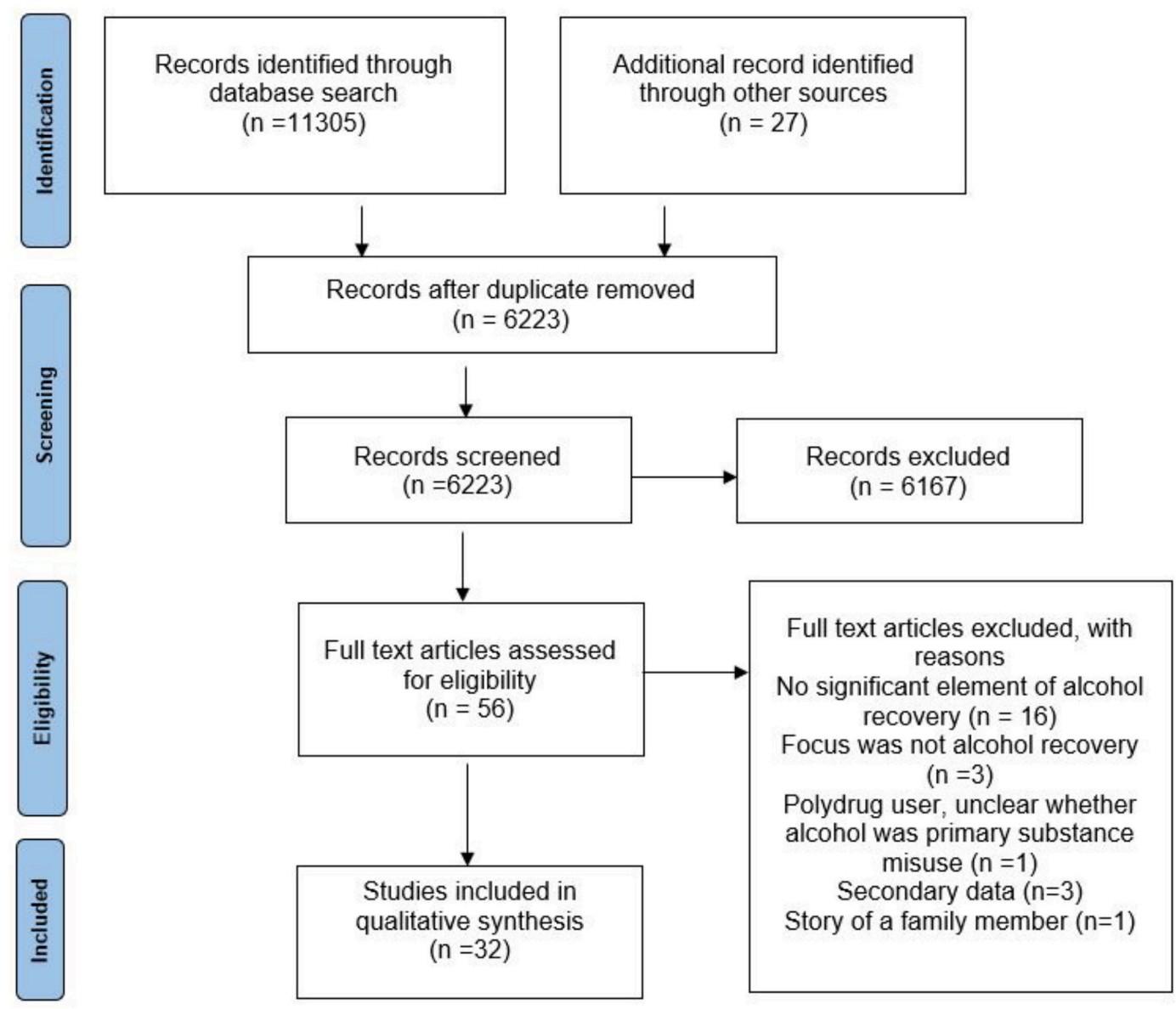

**Fig 1. PRISMA flow diagram for studies selection.**

documents were conducted in the United States (46.9% *n* = 15), followed by Europe (34.4%, *n* = 11). No included studies were from low-income countries. Of the included studies (*n* = 32), 29 used qualitative and 3 mixed methods (Table 1). The full references of included studies are provided in S2 Table.

## Quality assessment of included studies

Of the 32 studies, seven (21.9%) were rated high quality, 19 (59.4%) medium and six (18.8%) low quality (S3 Table).

**Table 1. Characteristics of included studies and participants.**

| Study ID | Lead Author | | Methods | | Participants | | | |
| | Academic discipline | Country | Setting of recovery | Study design, Data collection | Sample size (Male) | Age[e] | Ethnicity | Length of sobriety (years) |
| --- | --- | --- | --- | --- | --- | --- | --- | --- |
| Best et al., 2016 [7] | Social and health research | UK, USA | Glasgow addiction services | Quantitative, Structured interview | 205 M = 137) | 42 | - | 1–3 (n = 121) |
| | | | | | | | | 3–5 (n = 26) |
| | | | | | | | | >5 (n = 58) |
| Burman, 1997 [38] | Social Work | USA | Natural recovery[a] | Qualitative, Semi-structured interview | 38 (M = 24) | 22–73 | White = 34 | 1–26 |
| | | | | | | | Black = 3 | |
| | | | | | | | Other = 1 | |
| Cain, 1991 [27] | Anthropology | USA | Alcoholics Anonymous (AA) | Qualitative, Unstructured interview | 3 (M = 2) | - | - | 2–14 |
| | | | | | | | | Relapsed = 1 |
| Christensen and Elmeland, 2015 [26] | Psychology | Denmark | AA (11), Natural recovery (NR) (31) | Qualitative, Semi-structured interview | 42 (M = 26) | 45 | - | 2-10(AA) |
| | | | | | | | | 2-24(NR) |
| Dalgarno, 2018 [39] | Philosophy | Australia | Natural recovery, AA | Qualitative, Autobiographies | 7 | NA | Aboriginal | - |
| Dunlop and Tracy, 2013 [40] | Psychology | Canada | AA | Qualitative, Structured interview and questionnaire | 132 (M = 58) | 54, 38 | White = 99 | 0.3–4 |
| Dunlop and Tracy, 2013 [41] | Psychology | Canada | AA | Qualitative, Autobiographies | 46 (M = 23) | 22–82 | White = 34 Indigenous = 6 Other = 6 | 0.3–39 |
| Garland et al., 2012 [42] | Social Work | USA | Mindfulness-Oriented Recovery Enhancement | Qualitative, Semi-structured interview | 18 (M = 14) | 40 | White = 7 | - |
| | | | | | | | Black = 11 | |
| Gubi and Marsden-Hughes, 2013 [43] | Counselling | UK | AA | Qualitative, Semi-structured interview | 8 (M = 4) | 51–84 | White = 8 | 17–48 |
| Haarni and Hautamäki, 2010 [44] | Sociology | Finland | No specific treatment setting[b] | Qualitative, Semi-structured interview | 31 (M = 15) | 60–75 | - | Current and ex-consumer |
| Hanninen and Koski-Jannes, 1999 [11] | Social Psychology | Finland | Natural recovery, Therapeutic and self-help groups, AA, Psychiatrist consultation | Qualitative, Story writing by participants in 3rd person | 51 (M = 22) | - | - | - |
| Inman and Kornegay, 2004 [45] | Social Work | USA | Psychology clinics, medical rehabilitation groups, AA, Self-motivation | Qualitative, Semi-structured interview | 5 (M = 5) | 52–75 | - | 6-25(n = 3) |
| | | | | | | | | still drinking (n = 1) Controlled drinking (n = 1) |
| Jones, 2013 [46] | Sports Psychology | UK | Community alcohol services, AA, Sporting chance clinic | Qualitative, Open-ended interview | 1 (M = 1) | 30's | White | Sober |
| Laitman and Lederman, 2008 [47] | Substance abuse | USA | Rutgers college recovery support program | Qualitative, Un-specified | 1 (M = 0) | 19 | - | Sober |
| Laville, 2006 [48] | Community research | UK | Psychiatric unit, AKABA[c] | Qualitative, Self-narrative | 1 (M = 1) | 45 | Black | Sober |
| Lederman and Menegatos, 2011 [17] | Social sciences | USA | AA | Qualitative, Open-ended questionnaire | 178 (M = 86) | 19–75 | White = 171 | |
| Liezille Jacobs*, 2015 [49] | Public Health | South Africa | AA | Qualitative, Narrative interview | 10 (M = 0) | 30–62 | - | >0.6 |
| Mellor et al., 2021 [16] | Substance Misuse | Australia | Natural recovery | Qualitative, Semi-structured interview | 12 (M = 5) | 30–70 | - | No alcohol in 12 months (n = 6) |

*(Continued)*

**Table 1.** (Continued)

| Study ID | Lead Author | | Methods | | Participants | | | |
|---|---|---|---|---|---|---|---|---|
| | Academic discipline | Country | Setting of recovery | Study design, Data collection | Sample size (Male) | Age[e] | Ethnicity | Length of sobriety (years) |
| Mohatt et al., 2008 [50] | Psychology | USA | Natural recovery (38%), AA (33%), Combination of AA and other treatment programmes (29%) | Qualitative, Semi-structured interview | 57 (M = 26) | 26–72 | Alaskan Native | >5 |
| Newton, 2007 [51] | Adult liver transplant | USA | Liver transplant services | Mixed Methods, Unstructured interview | 76[f] (M = 39) | - | - | Relapsed = 4 |
| Opačić, 2019 [52] | Social Work | Croatia | Alcohol treatment services (n = 6), Natural recovery (n = 3) | Qualitative, Unstructured interview | 9 (M = 7) | 46–73 | - | 2–15 |
| Paris and Bradley, 2001 [53] | Psychology of recovery | USA | Natural recovery (2), AA (1) | Qualitative, Unstructured interview | 3 (M = 0) | 21–52 | - | 6–26 |
| Punzi and Tidefors, 2014 [54] | Psychology | Sweden | Alcohol residential care unit | Qualitative, Semi-structured interview | 5 (M = 4) | 50–60 | - | 0.8-several |
| Robbins, 2015 [55] | Nursing | USA | Alcohol treatment services | Mixed methods, Semi-structured interview | 21 (M = 0) | 37–67 | White = 15 Hispanic = 6 | 2 |
| Rowan and Butler, 2014 [56] | Social Work | USA | Natural recovery, AA, Alanon, ACOA[d] | Qualitative, Semi-structured interview | 20 (M = 0) | 50–70 | White = 19 B = 1 | 1–32 |
| Sawer et al., 2020 [57] | Psychology | UK | AA | Qualitative, Semi-structured interview | 8 (M = 5) | 27–74 | - | 1.9–35 |
| Stott and Priest, 2018 [58] | Clinical Psychology | UK | Substance misuse services, Specialist mental health services | Qualitative, Unstructured interview | 10 (M = 6) | 30–69 | White = 9 Black = 1 | Abstinent(n = 7), active (n = 3) |
| Strobbe and Kurtz, 2012 [59] | Psychiatry | USA | AA | Qualitative, Stories from AA "big book" | 24 (M = 14) | 17–75 | - | Sober |
| Suprina, 2006 [60] | Psychology | USA | AA | Mixed methods, BASIS-A Questionnaire, and Interview | 10 (M = 10) | 33–63 | White = 8 Black = 1 Latin = 1 | 3–25 |
| Vaughn and Long, 1999 [61] | Education | USA | AA | Qualitative, Semi-structured interview | 7 (M = 5) | 22–32 | White = 7 | 5–15 |
| Weegmann and Piwowoz-Hjort, 2009 [62] | Psychology | UK, Sweden | AA | Qualitative, Semi-structured interview | 9 (M = 4) | 40–75 | White = 9 | 9–23 |
| Zakrzewski and Hector, 2004 [63] | Psychology | USA | AA | Qualitative, Non-directive interviews | 7 (M = 7) | 32–65 | - | 1–25 |

The detailed reference list of included studies is provided in S2 Table.

[a]Natural recovery (recovery outside treatment setting,): The authors specified recovery outside treatment setting where; i) participant did not have formal alcohol treatment in an institution, organisation or by a person with an objective to relive alcohol problem. Or ii) No participation in substance abuse treatment or self-help groups 2 year prior to achieving abstinence or iii) Fewer than 9 sessions with AA or temperance society [16, 26, 38].

[b]No specific treatment settings: author did not specify settings.

[c]AKABA- Outreach support services for black men with mental health problems and substance misuse, run by Kush Supported Housing and Outreach services (98 Stoke Newington High Street, London, N167NY).

[d]ACOA-Adult children of alcoholics.

[e]Age in years is given as range or mean.

[f]Of all participants 18 had liver transplant for alcohol related liver disease.

## Participants

A total of 1055 participants were recruited across all included studies. The age range was 17–82 years, 52.1% (n = 550) of participants identified as male, 46.4% (n = 490) as female and 1.4% no gender specified (n = 15). Eight studies only included participants of a single gender. Only 16 studies accounting for 563 participants provided ethnicity details, 74.8% (n = 421) participants in these studies were white. Participants were recruited from various treatment settings; 12 studies solely recruited participants (41.9%, n = 442) known to AA, of these participants 49.3% (n = 218) were male. The length of sobriety of participants ranged from a few months to over three decades. Three studies [44, 45, 58] included both active and abstinent drinkers, in 1 study [16] half of participants had consumed alcohol in the past 12 months and 2 studies [27, 51] included participants who relapsed after a period of sobriety (Table 1).

## Conceptual framework

Eight dimensions (genre, identity, recovery setting, drinking trajectory, drinking behaviours and traits, stages, spirituality and religion, recovery experience) were derived and arranged in three superordinate categories: form, structure, and content. Each dimension had several types and subtypes, as specified in Table 2. The explanation and reference for individual dimensions is provided in S4 Table.

**1. Genre.**   Four genres from 13 studies were identified: **Drama; Redemption; Drinking tale, and Identity tale** (Table 3) [64, 65].

*Drama* has three subtypes. *Melodrama*: narratives that are high in emotional content and present exaggerated characters and exciting events. *Comedy theatre*: narratives with humorous element, which often use dramatic irony to induce laughter. *Quest*: narratives that take recipients on a journey in search of something (such as a successful recovery).

*Redemption* has two subtypes. *Redemptive narratives*: describes stories which centred on the idea of self-redemption, a phenomenon used to describe positive personal change after a

**Table 2. Dimensions of alcohol recovery narratives.**

| Superordinate category | Reference | Dimensions | Types | | | |
|---|---|---|---|---|---|---|
| Form | | | | | | |
| | [11, 17, 39, 40, 44, 45, 47, 49, 50, 55–58, 60, 61, 63] | **Genre** | Drama | Redemption | Drinking tale | Identity tale |
| | [7, 17, 26, 27, 38, 39, 43, 45, 53, 55, 57, 59, 61, 62] | **Identity** | Renewal | Construction | Formation | |
| | [7, 11, 16, 17, 26, 27, 38–63] | **Recovery setting (positioning)** | Recovery within treatment | Recovery outside treatment | | |
| Structure | | | | | | |
| | [7, 44] | **Drinking trajectory** | Upward | Fluctuating | Steady | Downward |
| | [27, 43, 52] | **Drinking behaviours and traits** | Non-alcoholic | Alcoholic | Personality traits | |
| | [7, 27, 42, 43, 45, 49, 50, 52, 56, 58, 59, 62, 63] | **Stages (sequence)** | Origin of difficulty | Episode of Change | Recovery | Ongoing struggle |
| Content | | | | | | |
| | [27, 38, 39, 53, 55, 56, 59–63] | **Spirituality and religion** | Religion versus spirituality | Belonging | | |
| | [38, 48, 51, 59] | **Recovery experience** | Positive | Negative | | |

The detailed reference list of included studies in provided in S2 Table.

**Table 3. Description of types and subtypes of alcohol recovery stories dimensions.**

| Genre | | | |
|---|---|---|---|
| *Drama* | *Redemption* | *Drinking tale* | *Identity tale* |
| Melodrama | Redemptive | Painful past | Stages of life |
| Comedy theatre | Non-redemptive | Reinforcement | Sex |
| Quest | | Loss of uniqueness | Sexual orientation |
| | | Relationship with oneself | Marginalised societies |
| | | Helping others | |
| **Identity** | | | |
| *Renewal* | *Construction* | *Formation* | |
| Motivation to change | Self-nurturing | Perceived Life change | |
| Emotional response | Beyond self | Adaptation | |
| Shame and crises | Cognitive restructuring | Acceptance | |
| Identity diffusion | Admittance and surrender | Reconstructing relationships | |
| | | Delivering back | |
| **Recovery setting (positioning)** | | | |
| *Recovery within treatment setting* | | *Recovery outside treatment setting* | |
| AA narratives | | Self-changer or natural recovery | |
| Dual diagnosis narratives | | Personal growth story | |
| Poly drug abuse narratives | | Emancipation narrative | |
| | | Discovery narratives | |
| | | Mastery narratives | |
| | | Coping narratives | |
| **Drinking trajectory** | | | |
| *Upward* | *Fluctuating* | *Steady* | *Downward* |
| Mildly upward drinking careers | Suspended drinking career | | Mildly downward drinking career |
| Sharply upward drinking career | | | Steeper downward drinking career |
| **Drinking behaviours** | | | |
| *Non-alcoholic* | | *Alcoholic* | *Alcohol impact* |
| Drinking | Nondrinking | Uncontrolled drinking | Antisocial |
| Controlled | Abstainer | Active alcoholic | Passive |
| Normal drinker | Nondrinking alcoholics | | Prosocial |
| Recovering alcoholic | Recovered alcoholics | | Grandiose |
| | | | Dishonest |
| **Stages (sequence) (can be non-linear)** | | | |
| *Origin of difficulty* | *Episode of Change* | *Recovery* | *Ongoing struggle* |
| Start of drinking | Blame and escape | Acknowledging problem | Being sober |
| Negative effect | Identification of problem | Surrender | Maintaining sobriety |
| Drinking progress | Alcoholic regression | Acceptance | Maintaining recovery |
| Problems | Rejection and denial | Help | |
| Drinking worsens | Turning points | Become sober | |
| **Spirituality and religion** | | | |
| *spirituality versus Religion* | | *Community Belonging* | |
| **Religion** | **Spirituality** | Lack of belonging | |
| Community | Individual | A search for belonging | |
| Bound | Limitless | Attain belonging | |
| Dogmatic and ritualistic | Flexible and transformative | | |
| Exclusive | Inclusive | | |
| **Recovery experience** | | | |

*(Continued)*

**Table 3.** (Continued)

| Positive | | Negative | |
|---|---|---|---|
| Ego ideal | | Craving | |
| Self-pride | | Intense self-discipline | |
| Empowerment | | Loss of drinking friends and social contacts | |
| Improved relationships | | Intrusive disturbing memories | |
| Improved trust in family | | Inadequate coping skills to face reality | |
| Reintegration into society | | Depression, anxiety | |
| Lost opportunities found | | Loneliness | |
| Happy to be alive | | Work and financial issue | |
| Enjoy doing thing | | Impact of comorbidities | |
| | | Life stinks | |

negative experience [40]. Redemptive narratives were often shared by narrators who were in long term recovery from alcohol misuse, and who perceived they had benefited from their adversities [66]. They showed elements of difficult experience, positive self-transformation, greater improvement in general health, and had a high chance of sustained sobriety. *Non-redemptive narratives*: had short term recovery, lacked positive experience, had less improvement in general health, and increased risk of relapse to drinking [40].

*Drinking Tale* describes how sharing a narrative impacted the narrators themselves [67]. Sharing life stories helped the narrator's recovery in five different ways; by being reminded of their painful past, reinforcing their own recovery, losing their sense of uniqueness, facilitating and improving their relationship with themselves, and eventually helping others [17].

*Identity Tale* comprised narratives which foregrounded characteristics in relation to their alcohol use and social context (e.g., narrator's age, gender, sexual orientation, ethnicity). Some research specifically sought the narratives of marginalised people such as Indigenous Australians and Alaskans. Drinking behaviour and recovery varied by life stage. Associated characteristics expounded through subgroup analysis.

**2. Identity.** Identity as a dimension describes self-transformation as a multistage process, distinct from the use 'identity tale' whereby the later highlights social, cultural, and demographic aspects. Fourteen publications discussed the importance of identity in the context of alcohol recovery. The concept of identity acquisition is a cornerstone of recovery in AA, where a person who has problems with alcohol accepts "alcoholism" as a disease and identifies as an "alcoholic" [27]. This concept of identity acquisition is not generally used in recovery outside formal treatment settings [16, 38].

The concept of identity transformation was characteristic of these narratives., Within this dimension, we identified following stages—**identity renewal**, **identity construction**, and **identity formation** (Table 3).

*Identity renewal.* During this first stage, the individual lacks a specific identity, nor is effort expended in forming one, a phenomenon described in psychological literature as "identity diffusion" [68]. Alcohol misuse causes a personal and social crisis and the person experiences fear, guilt, and shame. Participants spoke of recuperating and rebounding from "rock bottom".

*Identity construction.* The ensuing stage comprises of self-nurturing where a person arrives at a point where they begin to look for help, share their situation with others and 'surrender' to the process of recovery from alcohol misuse [59, 61, 62]. The individual goes through cognitive restructuring, whereby one starts giving up on destructive thoughts, believing in the self, commits to change and attains a new identity [38].

*Identity formation*. In the final stage a person accepts their renewed identity as a self-aware "alcoholic". What followed in the narratives was affinity and group membership, adapting to their emerging new role. The narratives characterised reconstructing social identity and mending relationships and generating capacity to help others [11, 27, 53].

**3. Recovery setting.** In recovery setting type two subtypes were identified 'recovery within treatment' and 'recovery outside treatment' (Table 2). 'Recovery within treatment' describes the experiences of a participant who was formally treated by an institution, clinicians, alcohol support workers, organisation, or a person for alcohol misuse. 'Recovery outside treatment' describes the experiences a participant who had minimal or no formal input from an institution, clinicians, alcohol support workers, organisation, or a person for alcohol misuse [16, 26, 38].

*Recovery within treatment* has following subtypes. *AA narrative*: was most common for recovery within a formal treatment system, the core of an AA narrative was hitting rock bottom, sharing a story, spirituality, and acceptance of the new identity as an"alcoholic" [11, 26, 27, 45, 55]. *Dual diagnosis*: has narratives of *alcohol misuse and mental health problems*, and *alcohol misuse and diabetes* [45, 58].

*Alcohol misuse and mental health* has the following narratives. *Dominant cultural narrative*: participants were more inclined to accept a diagnosis of a mental health problem but were resistant to the label of an "alcoholic". *Community and family narratives*: participants described recover as an ongoing process involving significant others and achieving recovery by a sense of belonging, mutual aid, and sharing experiences. In both contexts mental health services played a pivotal role in recovery processes [58].

*Alcohol misuse and diabetes*: In these narratives all participants believed in the genetic inheritance of diabetes but not of "alcoholism". Participants often confused symptoms of alcohol withdrawal with hypoglycaemia which resulted in erratic eating and drinking habits. The involvement of specialist diabetic services and alcohol support groups improved participant knowledge and facilitated recovery [45].

*Polydrug misuse* has narratives of participants who suffered childhood trauma, a strict code of keeping family secrets and denying negative feelings, resulting in multiple substances addiction. Therapeutic and self-help groups played an important role in recovery of people with these experiences [45].

*Recovery outside a treatment* has following subtypes. *Natural recovery*: narratives were less homogenous than those within treatment setting. They included internal and external influences, did not feature significant involvement of others. Participants who described natural recovery tended to disagree with labelling and did not believe sharing stories helped recovery [26, 38]. Cognitive restructuring and positive recovery capital played a key role in natural recovery [7, 38]. *Emancipation narratives*: described identity development through making changes in life and liberation from oppressive circumstances. *Discovery narratives*: in these narratives participants identified themselves being different and developed their identity by consciously expanding experiences including art and the use of psychedelic drugs such as LSD. *Mastery narratives*: in these narratives' participants felt social pressure to demonstrate mastery over things like to win fights and/or drink more, alcohol misuse was seen as irrational behaviour, with recovery involving an increased awareness of a drinking problem. *Coping narratives*: described a lifelong struggle, difficult personal circumstances, and use diagnostic labels to help recovery [16].

**4. Drinking trajectory.** The drinking trajectory describes impact of aging on drinking habits and comprises of four types (Table 3) [7, 44].

*Upward drinking career* describes the increase of alcohol intake in adulthood and had two further subtypes 'mildly upward' and 'sharply upward'. In the '*mildly upward*' career alcohol

was part of social life and slowly increased with age. The '*sharply upward*' drinking trajectory found to be common in women, with drinking becoming part of the person's lifestyle in the later part of their working years.

*Fluctuating drinking career* describes drinking patterns which varied with time and life circumstances.

*Steady drinking career* describes intermittent periods of sobriety and heavy alcohol use.

*Downward drinking career* describes decline in alcohol consumption as the person got older. This was either *mildly downward*, where change was slow, or *steeply downward*, where change was rapid [44]. Alcohol careers can include late onset of alcohol dependence [often after specific triggers such as bereavement or retirement] with resolution shortly thereafter [7]. Dunlop et al. (2013) showed age positively correlates with improved self-esteem, general health, and authentic pride and negatively with aggression which in turn increase the chances of recovery from alcohol misuse [41].

**5. Drinking behaviours and traits.**   *Non-alcoholic drinking* type comprises narratives participants were drinking actively but in a controlled manner.

*Non-alcoholic non-drinking* type comprises narratives of participants who completely abstained from alcohol. In '*alcoholic drinking*' type participants were active alcoholics. In '*alcohol non-drinking*' type the participants were either 'non-drinking alcoholics' or 'recovering alcoholics' [27, 44].

*Personality traits* including antisocial, passive, prosocial, grandiose, and dishonest were commonly associated with alcohol misuse [43, 52].

**6. Stages (sequence).**   The commonly used alcohol recovery model has the following stages: origin of difficulty, episode of change, attainment of recovery, and ongoing struggle (Table 3) [43, 50, 58]. In these narratives triggers of alcohol use were social and cultural difficulties, norms and pressures, childhood abuse, mental health problems, a lack of belonging and numbing the pain [47, 60, 61]. As drinking progressed, physical, mental health, and social problems attributable to alcohol consumption developed, with alcohol escalating to provide escape from fear and shame. Turning points described by participants ranged from no specific event to near death experiences, embarrassment, spiritual experiences, a sense of loss, death of a family member, loss of a friend by suicide, and physical and mental health decline [56, 63]. The person described a phase of rejection and denial, but eventual acknowledgement of the problem followed by help seeking or natural recovery, then sobriety. Ongoing struggle describes the efforts made by the individual to maintain their sobriety and recovery [43]. By participating in meaningful activities, adopting a new identity, and creating positive recovery capital narrators of these stories felt they were more likely to achieve long term sobriety [7].

**7. Spirituality and religion.**   A lack of sense of belonging was a common theme that resonated across numerous recovery stories, and particularly in stories from more marginalised communities such as Indigenous Americans and Australians and those in the LGBTQ+ community [39, 50, 56, 60]. Spirituality and belief in a higher power was a cornerstone for recovery in the AA model [27, 45]. Participants described **'religion'** as dogmatic, ritualistic, biased against sexual orientation and identity, and had strict codes of moral behaviour, while **'spirituality'** as more individualistic, open, inclusive, and flexible [60]. Lack of belonging and social isolation triggered alcohol use, and support groups such as AA provided an opportunity for spiritual reconnection and f attainment of a sense of belonging and sobriety [53, 56, 59, 60].

**8. Recovery experience.**   Recovery experience narratives were positive, negative or both (Table 3).

*Positive recovery experiences* were ego ideal for participants, and improved their self-pride, empowerment, trust, and relationships. They found lost opportunities, felt more integrated into society, were happy to be alive, and enjoyed new hobbies and activities.

*Negative recovery experiences* were characterised as having a craving for alcohol, feeling the pressure of intense self-discipline, loosing drinking friends and social contacts, inadequate coping skills, and concomitant mental health illness. The narratives liver transplants recipients particularly offered the themes of financial and job-related issues and the impact of other comorbidities [38, 51].

**Subgroup analysis.** *Age*. Along the dimension of age, young people, drinking habits and activities often involved peer pressure whilst socialising with friends, such as taking part in drinking games in college and as part of social status, whereas drinking habits of older individuals related to later life experiences and challenges [44, 47]. Thus, demonstrating an importance of social and cultural influences on drinking behaviours, which may influence recovery [64, 65].

*Gender*. Five studies reported the narratives of female participants only, these studies emphasized identity renewal, and the affective response of shame as characteristic of the recovery narrative [47, 49, 53, 55, 56]. Shame is a social and regulatory emotion that invokes self-awareness and self-other obligations [49, 56]; we also found an all-female study using shame as impetus to build relationships through help of networks. This was a common affective response that contributed toward coping when stepping out of addiction and into new identity. There was a heavy reliance on social networks, which was present in all narratives apart from Christensen and Elmeland (2015) where participants used new hobbies and activities for self-renewal. Studies with male only participants showed no distinct characteristics in the sample except the study using shame as impetus described above [26].

*Sexual orientation*. In the studies with participants identifying themselves as LGBTQ+, we note that spiritual awakening was more commonly sought rather than religious affiliation [56, 60]. Alcohol use was a lifestyle choice recognised by participants from the LGBTQ+ community. Building a new identity through recovery programs and networks enabled recovery and formation of new 'productive' relationships outside of alcohol use [56, 60].

*Marginalised communities*. Analysis of studies discussing the experiences of Indigenous Australian and Alaskan people's recovery [39, 50] although showings experience of similar stages of recovery, tended to have more emphasis on elements of stereotyping, alienation, marginalisation, inequality, low wages, and the impact of sudden gaining of citizenship status and money. The recovery process was unpredictable and messy [39], and participants achieved recovery both within and outside treatments settings.

*Alcohol and mental health*. Analysis of studies discussing dual diagnosis of alcohol misuse and mental health problems showed participants often suffered with negative self-perceptions, including low self-esteem, lack of love from others, lack of desire to belong, anger, and shame [11, 38, 40, 43, 46, 53, 58, 59, 61]. Mental health problems often acted as a trigger to drink harmfully [61]. Common mental health problems reported were anxiety, depression, obsessive compulsive disorders, post traumatic disorders [mostly due to difficult childhoods], attention seeking behaviours, eating disorders, and emotional instability [11, 40, 43, 46, 58, 61]. Facilitators to recovery were integrated support from mental health and substance misuse services, a flexible and trustworthy relationship with care providers, individualised treatment pathways, and easy to understand step-by-step support. Whereas barriers to recovery were undiagnosed or unrecognised mental health problems, inadequate support from mental health services, underfunded services, and punitive response to alcohol misuse [58]. Participants who had negative recovery experience reported ongoing mental health difficulties comprised of anxiety, depression, and intrusive or disturbing memories. This in turn impacted longevity of sobriety [38].

*Medium and high-quality studies*. On performing subgroup analysis on twenty-six medium and high-quality studies most dimension types were present in the framework apart from the narratives of college drinking, indigenous Australians, and redemption.

## Discussion

The current review identified a rich source of existing literature describing alcohol recovery narratives and summarised identified characteristics. Included studies were multi-disciplinary and summarised alcohol recovery experiences of over a thousand participants spanning 30 years of research. Narratives analysed in included studies belonged to people from a variety of social and demographic orientations. Although this sample was not entirely diverse in term of ethnic distribution, the review does include studies which voiced recovery experience of more marginalised communities such as Alaskans and indigenous Australians [39, 50]. The review collated a diverse source of multidimensional narratives using conceptual similarities and differences into eight dimensions with each its own specific types and subtypes. This conceptual framework provides researchers, practitioners, policy makers and others with an accessible resource to build future research and practice.

Our review demonstrated the dynamic nature of recovery as a nonlinear- and non-dichotomous process, which supports previous work [28]. The subtype 'ongoing struggle' was important for giving voice to some people's continued daily efforts to recover. Our work highlighted the diversity in participants narratives based on multiple factors such as recovery setting, age, gender, sexual orientation, and ethnicity. Participants recovered from alcohol misuse both within and outside formal treatment settings, however the majority of included studies described achieving recovery through AA or participants who interacted with more than one service and tried numerous recovery strategies [28]. In our review, 41% of participant narratives were from people who were known to AA. A Cochrane review found AA and other 12-Step programmes were superior to other clinical interventions at continuous abstinence from alcohol both in the short and long term. However, the authors acknowledge that those who do not see improvements of AA after a certain period should be offered a different approach [69]. Narratives from people who had followed the AA model in our review used similar types of language e.g., 'rock bottom'. Those who rejected formal treatment of this kind and opted for 'natural recovery' described not being able to relate to the language and concepts used in AA. Our work may help better understand the characteristics of those who find AA works for them, and those who do not, which would reduce uptake of multiple treatment modalities and feelings of frustration.

The genres we identified characterize recovery narratives in four ways. These are drama, redemption, drinking tale and identity tale, which in different ways demonstrate a progression of an emotional self, actively constructing an identity to aid stepping out of addictive lifestyle practices. We found stages of identity construction were representative of reviewed narratives of alcohol recovery. The individuals grow through identity renewal, identity construction and identity formation to often find sustainable recovery, sometimes finding themselves in a role to help others struggling with addiction [70]. The motivation to reinvent the self by construction of new identity is a behavioural patterns associated with addiction [71]. The stages we observed uses narratives to demonstrate the argument that recovery is largely driven by a personal and affective evaluation of the self, leaving behind one identity in pursuit of another [7]. That is, the individual returns, when useful in the narrative, to a mode of evaluation regarding how bad things are (current identity) and how reachable and better things could be (renewed identity within the new group) [72].

### Strengths and limitations

The following strengths of the review noted. First, the review has a comprehensive search strategy, piloted, and finalised in consultation with a senior librarian. Second, the review team was

consisted of multidisciplinary members with diverse experiences and including people with experience of alcohol misuse. This enabled rich discussion among review team and careful consideration while choosing terms to describe alcohol misuse, social context of participants including sexual orientation. Third, a three-stage data synthesis approach was adopted to achieve robustness of process.

The following limitations of the review were noted. First, the results of the review may not be generalised to low-income countries, and non-Caucasian populations as all the included studies were conducted in high income countries with white predominant population. Detailed ethnic distribution was missing in most studies and the search strategy was restricted to the English language. Second, author's personal viewpoints and experiences might have influenced the date interpretations, to minimise this, we followed three stage approach for data synthesis. Finally, as the focus of the review was to explore recovery from a primary problem of alcohol misuse, it was beyond the scope to examine polydrug use in detail. Future reviews may wish to focus explicitly on this complexity.

## Implications for research and practice

We contribute an understanding of narratives in relation to both structured support and unsupported 'natural' journeys of recovery; an area that remains poorly developed and understood in research [16] and we recommend should be expanded. Our study assimilates types of narratives recognised in the literature such as emancipation, discovery and mastery, and contributes the distinction of unstructured recovery narratives as cognitively loaded (i.e. mental effort in restructuring beliefs and coping with associated emotions), involving meaningful activity like art and psychedelic drugs, and with less involvement and support from others [16, 70]. Our review finds evidence through narratives of recovery from alcohol, for the notion of recovery as motivated by push factors (hitting rock bottom, shame, identity loss, alienation) and pull factors (the good life, the social relationships one wants to develop and starts to enjoy) [73]. This dynamic applied to individuals from a range of social orientation, actively seeking renewal of identity.

We found that the path to recovery involved some higher order (religious/spiritual) system of thought and practice toward what is more broadly recognised in addiction research as the recoveree "developing a sense of future" [74]. Driven emotionally with hope and positive feelings, individuals found forming or mending relationships with significant others helped their recovery. Through meaningful activity, they acquired goals, acquired safety and confidence, often in a program that offered a social support network. We note that amongst individuals who were part of the LGBTQ+ community, recovery from alcohol misuse was particularly aided by a sense of belonging to groups. Latent mental health problems were described as acting as a trigger in some narratives, and narratives describing dual diagnoses provided information about forms of mental health intervention that helped (including effective services) and did not help (including pejorative treatment of alcohol use).

## Conclusion

The role of narratives in alcohol recovery is only partially understood [59, 75]. In this context, our review provides characteristics of alcohol recovery narratives, with implications for both research and healthcare practice. We recommend research focus on collecting narratives from people in lower income countries, in those who have recovered outside of mainstream services or those who have used services other than AA, with a focus on more ethnic diversity in studies.

## Supporting information

**S1 Table. Sample search strategy for Ovid Medline.**
(XLSX)

**S2 Table. Full reference list of included studies.**
(XLSX)

**S3 Table. Risk of bias and quality of included studies.**
(XLSX)

**S4 Table. Alcohol recovery narrative dimensions, references, and definitions.**
(XLSX)

**S1 Data.**
(XLSX)

## Acknowledgments

Alison Ashmore, senior research librarian (Nottingham University Libraries) contributed to finalising the search strategy.

## Author Contributions

**Conceptualization:** Mohsan Subhani, Joy Llewellyn-Beardsley, Stefan Rennick-Egglestone.

**Data curation:** Mohsan Subhani, Usman Talat, Holly Knight, Stefan Rennick-Egglestone.

**Formal analysis:** Mohsan Subhani, Usman Talat, Holly Knight.

**Methodology:** Mohsan Subhani, Usman Talat, Holly Knight, Katy A. Jones, Joy Llewellyn-Beardsley, Stefan Rennick-Egglestone.

**Project administration:** Mohsan Subhani.

**Resources:** Joy Llewellyn-Beardsley, Stefan Rennick-Egglestone.

**Software:** Mohsan Subhani.

**Supervision:** Stefan Rennick-Egglestone.

**Writing – original draft:** Mohsan Subhani.

**Writing – review & editing:** Mohsan Subhani, Usman Talat, Holly Knight, Joanne R. Morling, Katy A. Jones, Guruprasad P. Aithal, Stephen D. Ryder, Joy Llewellyn-Beardsley, Stefan Rennick-Egglestone.

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
