## [Decision Letter · Decision Letter 0]

22 Mar 2022

PONE-D-21-32823Characteristics of alcohol recovery narratives: systematic review and narrative synthesisPLOS ONE

Dear Dr. Subhani

Thank you for submitting your manuscript to PLOS ONE. After careful consideration, we feel that it has merit but does not fully meet PLOS ONE’s publication criteria as it currently stands. Therefore, we invite you to submit a revised version of the manuscript that addresses the points raised during the review process.

We look forward to receiving your revised manuscript.

Kind regards,

Saeed Ahmed, MD

Academic Editor

PLOS ONE

Journal Requirements:

The review was funded by National Institute for Health Research as part of feasibility randomised control trial titles “Does knowledge of liver fibrosis affect high risk drinking behaviour (KLIFAD)? A feasibility randomised controlled trial”. Funding award ID: NIHR201146. JRM and HK receive salary support from a Medical Research Council Clinician Scientist Fellowship [grant number MR/P008348/1]

The funders had no role in study design, data collection and analysis, decision to publish, or preparation of the manuscript

NO authors have competing interests

6. Please include your tables as part of your main manuscript and remove the individual files. Please note that supplementary tables (should remain/ be uploaded) as separate "supporting information" files.

Reviewers' comments:

Review Comments to the Author

**Reviewer #1:** This review is informative and helps the reader gain an easy understanding about the process of alcohol recovery, as well as the importance of incorporating narratives.

It may be helpful if the authors can add more details about the quality analysis of the studies reviewed. While the use if critical appraisal skills program has been mentioned, for the reader who may not be aware of this tool, a few additional details will help give a clearer picture as to how the quality of the study was assessed.

**Reviewer #2: **This systematic review is an excellent and balanced overview that includes 32 studies (29 qualitative, 3 mixed-methods, 1055 sample size from Europe and United states, includes multiple databases, spanning 30 years (male predominant, 41 % from AA recovery group), gives an interesting perspective of role of recovery narratives in alcohol recovery. It aims to produce a conceptual framework with multiple dimensions describing the characteristics of alcohol misuse recovery narratives from the available literature. It focuses mostly on Alcoholic Anonymous(AA) or other unstructured formal treatment for alcohol use recovery. It has a good subgroup analysis of all dimensions. Major limitation is, its missing people in lower income countries, as no research literature available in those areas, lacks information who have recovered outside of main services or those who have used services other than AA, and lacks data on females population and ethnic diversity especially in non Caucasian. It does include indigenous population and LGBTQ. Furthermore, this framework will help to identify gaps in knowledge, summarize range of methods, enables developing future research.

They have registered in Prospero, followed Prisma guidelines, is part of NIH grant, did CASP for ROB assessment, Has good Inclusion, exclusion criteria, and approach towards outcomes.

However, I have a suggestions that could be added to the article.

In my opinion, Tabulating all of these dimension definitions in a excel sheet with listing the articles or adding this information to the attached excel sheet would Make it more easy to follow. It will help to improve the flow and readability of the text. If a further subgroup analysis could be added separately to include information on concomitant substance use disorders(drug use) and/or mental health issues, then it would give a good perspective in terms forming a better framework. I recommend the article be published.

**Reviewer #4:** Nicely written article which identifies a conceptual framework including various aspects of chronic alcohol abuse and dependence, the narrative summarize the characteristics well, more than 1000 patient included and psychosocial aspects were explored which adds value

**Reviewer #6: **This paper aims to produce a conceptual framework describing the characteristics of alcohol misuse recovery narratives that are in the research literature, to inform the development of research, policy, and practice. The systematic review and data synthesis were well organized and all the elements needed for this type of quantitative research appear to be in place. Much of the tabular information is well summarized in the appendices.

The review may have provided characteristics of alcohol recovery narratives, with implications for both research and healthcare practice. It demonstrated knowledge gaps in relation to alcohol recovery narratives of people living in lower income countries, or those who recovered outside

of mainstream services.

Although the information is descriptive and quantitative, the risk of bias from each study should have been represented in tabular form. This is lacking in Table 1 of the supplemental material. Some comments concerning possible sources of heterogeneity and publication bias should have been addressed, especially with the geographical distribution of the data.

PLOS authors have the option to publish the peer review history of their article (what does this mean?). If published, this will include your full peer review and any attached files.

---

## [Author Response · Author response to Decision Letter 0]

30 Mar 2022

Dr Saeed Ahmed, MD

Academic Editor

PLOS ONE

Date: 25/03/2022

Dear Dr Ahmed, 

I am writing to address the suggested revisions to our submitted manuscript titled, "Characteristics of alcohol recovery narratives: systematic review and narrative synthesis". Thank you to reviewers for taking the time to provide a comprehensive review. 

We have attempted to address editor’s and each of the reviewer's comments, as listed below:

Editor:

Thank you very much for your comment. We have now revised the manuscript style and file names as per PLOS ONE guide to make it compliant with PLOS ONE's style requirements. 

Thank you very much for your comment. Apologies for an oversight we have now updated the grant information and have provided with the correct details. 

“The review was funded by National Institute for Health Research as part of feasibility randomised control trial under scheme Research for Patient Benefit (RfPB). Funding award ID: NIHR201146. The funders had no role in study design, data collection and analysis, decision to publish, or preparation of the manuscript.”

The review was funded by National Institute for Health Research as part of feasibility randomised control trial titles “Does knowledge of liver fibrosis affect high risk drinking behaviour (KLIFAD)? A feasibility randomised controlled trial”. Funding award ID: NIHR201146. JRM and HK receive salary support from a Medical Research Council Clinician Scientist Fellowship [grant number MR/P008348/1]

The funders had no role in study design, data collection and analysis, decision to publish, or preparation of the manuscript

Thank you very much for your comment. We have now removed funding information from acknowledgment section and from manuscript. 

Please add following statement in funding section of online submission form.

“The review was funded by National Institute for Health Research as part of feasibility randomised control trial under scheme Research for Patient Benefit (RfPB). Funding award ID: NIHR201146. The funders had no role in study design, data collection and analysis, decision to publish, or preparation of the manuscript.”

Please add following statement in acknowledgement section of online submission form.

“Alison Ashmore, senior research librarian (Nottingham University Libraries) contributed to finalising the search strategy.”

NO authors have competing interests

Thank you very much for your comment. We will be grateful for updating the online submission form on our behalf. Please add following statement

Thank you very much for your comment. Thanks for highlighting the issue. We have now removed these figures from supplementary material. 

6. Please include your tables as part of your main manuscript and remove the individual files. Please note that supplementary tables (should remain/ be uploaded) as separate "supporting information" files.

Thank you very much for your comment. We have now provided the tables as part of main manuscript and have uploaded supplementary material as separate files titled " S1 Table, S2 Table, S3 Table, and S4 Table "

Reviewers' comments:

Review Comments to the Author

Reviewer #1: 

This review is informative and helps the reader gain an easy understanding about the process of alcohol recovery, as well as the importance of incorporating narratives.

Thank you very much for your comment. We hope the article will attract similar attention from the wider audience of the journal.

It may be helpful if the authors can add more details about the quality analysis of the studies reviewed. While the use if critical appraisal skills program has been mentioned, for the reader who may not be aware of this tool, a few additional details will help give a clearer picture as to how the quality of the study was assessed.

Thank you very much for your comment. We have now provided details on quality analysis of included studies as advised by the reviewer. We have included following paragraph in method section of study. 

“Quality assessment of qualitative evidence synthesis has been a matter of debate for many decades (1). Cochrane Qualitative and Implementation Methods Group recommendations are to use a tool that takes the multi-dimensional concept of qualitative evidence into account (1). 

Keeping this in view, the quality of included studies and risk of bias was assessed using the Critical Appraisals Skills Programme (CASP) tool for qualitative research (2). The CASP tool focuses on three domains, study design, results validity, and generalisability. Each domain has a set of questions. Based on the response to these questions the studies were marked as low, medium, or high quality. The studies which provided satisfactory information in all domains were marked as high quality, with missing or unsatisfactory information in one domain as medium quality, and with missing or unsatisfactory information in two or more domains as low quality.”

Reviewer #2: 

This systematic review is an excellent and balanced overview that includes 32 studies (29 qualitative, 3 mixed-methods, 1055 sample size from Europe and United states, includes multiple databases, spanning 30 years (male predominant, 41 % from AA recovery group), gives an interesting perspective of role of recovery narratives in alcohol recovery. It aims to produce a conceptual framework with multiple dimensions describing the characteristics of alcohol misuse recovery narratives from the available literature. It focuses mostly on Alcoholic Anonymous(AA) or other unstructured formal treatment for alcohol use recovery. It has a good subgroup analysis of all dimensions. Major limitation is, its missing people in lower income countries, as no research literature available in those areas, lacks information who have recovered outside of main services or those who have used services other than AA, and lacks data on females population and ethnic diversity especially in non Caucasian. It does include indigenous population and LGBTQ. Furthermore, this framework will help to identify gaps in knowledge, summarize range of methods, enables developing future research.

They have registered in Prospero, followed Prisma guidelines, is part of NIH grant, did CASP for ROB assessment, Has good Inclusion, exclusion criteria, and approach towards outcomes.

Thank you very much for your appreciation and taking time to review the manuscript. 

However, I have a suggestions that could be added to the article.

In my opinion, Tabulating all of these dimension definitions in a excel sheet with listing the articles or adding this information to the attached excel sheet would Make it more easy to follow. It will help to improve the flow and readability of the text. 

Thank you very much for your comment. The detail of individual references with study ID and reference number has been provided in Supporting S2 Table. We have now added S4 Table in supporting information including the references and explanation of individual dimensions. 

If a further subgroup analysis could be added separately to include information on concomitant substance use disorders(drug use) and/or mental health issues, then it would give a good perspective in terms forming a better framework. I recommend the article be published.

Thank you very much for your comment. We have now added additional subgroup analysis describing dual diagnosis aspect of alcohol misuse and mental health. The data was insufficient to have sperate subgroup analysis on polydrug use. We have added this as a limitation of the review. The additional subgroup analysis reads as

Alcohol and mental health: Analysis of studies discussing dual diagnosis of alcohol misuse and mental health problems showed participants often suffered with negative self-perceptions, including low self-esteem, lack of love from others, lack of desire to belong, anger, and shame (3-11). Mental health problems often acted as a trigger to drink harmfully (5). Common mental health problems reported were anxiety, depression, obsessive compulsive disorders, post traumatic disorders (mostly due to difficult childhoods), attention seeking behaviours, eating disorders, and emotional instability (3-7, 9). Facilitators to recovery were integrated support from mental health and substance misuse services, a flexible and trustworthy relationship with care providers, individualised treatment pathways, and easy to understand step-by-step support. Whereas barriers to recovery were undiagnosed or unrecognised mental health problems, inadequate support from mental health services, underfunded services, and punitive response to alcohol misuse (9). Participants who had negative recovery experience reported ongoing mental health difficulties comprised of anxiety, depression, and intrusive or disturbing memories. This in turn impacted longevity of sobriety (11). 

Reviewer #4: 

Nicely written article which identifies a conceptual framework including various aspects of chronic alcohol abuse and dependence, the narrative summarize the characteristics well, more than 1000 patient included and psychosocial aspects were explored which adds value.

Thank you very much for your comment. We hope the article will attract similar attention from the wider audience of the journal. 

Reviewer #6: 

This paper aims to produce a conceptual framework describing the characteristics of alcohol misuse recovery narratives that are in the research literature, to inform the development of research, policy, and practice. The systematic review and data synthesis were well organized and all the elements needed for this type of quantitative research appear to be in place. Much of the tabular information is well summarized in the appendices.

The review may have provided characteristics of alcohol recovery narratives, with implications for both research and healthcare practice. It demonstrated knowledge gaps in relation to alcohol recovery narratives of people living in lower income countries, or those who recovered outside

of mainstream services.

Thank you very much for your comment and providing the feedback. 

Although the information is descriptive and quantitative, the risk of bias from each study should have been represented in tabular form. This is lacking in Table 1 of the supplemental material. Some comments concerning possible sources of heterogeneity and publication bias should have been addressed, especially with the geographical distribution of the data.

Thank you very much for your comment. Sorry for not providing the specific details, we have now added supporting S3 table describing risk of bias and quality assessment for individual studies. 

For reviewer convenience we have also included table at end of this document. 

Thank you again for your time and consideration.

Yours Sincerely,

Dr Mohsan Subhani

MBBS, MRCP Medicine, MRCP Gastroenterology

Nottingham Digestive Diseases Biomedical Research Centre (NDDC)

University of Nottingham UK

---

## [Decision Letter · Decision Letter 1]

21 Apr 2022

Characteristics of alcohol recovery narratives: systematic review and narrative synthesis

PONE-D-21-32823R1

Dear Dr. Subhani,

We’re pleased to inform you that your manuscript has been judged scientifically suitable for publication and will be formally accepted for publication once it meets all outstanding technical requirements.

Kind regards,

Saeed Ahmed, MD

Academic Editor

PLOS ONE

**Comments to the Author**

1. If the authors have adequately addressed your comments raised in a previous round of review and you feel that this manuscript is now acceptable for publication, you may indicate that here to bypass the “Comments to the Author” section, enter your conflict of interest statement in the “Confidential to Editor” section, and submit your "Accept" recommendation.

Reviewer #2: All comments have been addressed

Reviewer #3: All comments have been addressed

Reviewer #4: All comments have been addressed

2. Is the manuscript technically sound, and do the data support the conclusions?

Reviewer #2: Yes

Reviewer #3: (No Response)

Reviewer #4: Yes

3. Has the statistical analysis been performed appropriately and rigorously? 

Reviewer #2: N/A

Reviewer #3: (No Response)

Reviewer #4: (No Response)

4. Have the authors made all data underlying the findings in their manuscript fully available?

Reviewer #2: Yes

Reviewer #3: (No Response)

Reviewer #4: Yes

5. Is the manuscript presented in an intelligible fashion and written in standard English?

Reviewer #2: Yes

Reviewer #3: (No Response)

Reviewer #4: Yes

6. Review Comments to the Author

Reviewer #2: well writing and good frame work

ROB included

Sub group analysis included

Good reading flow

supporting material included

Reviewer #3: (No Response)

Reviewer #4: Good review that provides characteristics of various alcohol recovery narratives. It includes aspects of research and healthcare practice. Managed to elicit knowledge gaps in relation to alcohol recovery narratives of consumers in underprivileged/ low education/ lower income countries, particularly those who recovered outside standard healthcare structures.

7. PLOS authors have the option to publish the peer review history of their article (what does this mean?). If published, this will include your full peer review and any attached files.

Reviewer #2: No

Reviewer #3: No

Reviewer #4: No

---

## [Editor Report · Acceptance letter]

25 Apr 2022

PONE-D-21-32823R1 

Characteristics of alcohol recovery narratives: systematic review and narrative synthesis 

Dear Dr. Subhani:

I'm pleased to inform you that your manuscript has been deemed suitable for publication in PLOS ONE. Congratulations! Your manuscript is now with our production department. 

Kind regards, 

on behalf of

Dr. Saeed Ahmed 

Academic Editor

PLOS ONE